# QueryMatch: A Query-based Contrastive Learning Framework for Weakly Supervised Visual Grounding

## ABSTRACT

Visual grounding is a task of locating the object referred by a natural language description. To reduce annotation costs, recent researchers are devoted into one-stage weakly supervised methods for visual grounding, which typically adopt the anchor-text matching paradigm. Despite the efficiency, we identify that anchor representations are often noisy and insufficient to describe object information, which inevitably hinders the vision-language alignments. In this paper, we propose a novel query-based one-stage framework for weakly supervised visual grounding, namely QueryMatch. Different from previous work, QueryMatch represents candidate objects with a set of query features, which inherently establish accurate one-to-one associations with visual objects. In this case, QueryMatch re-formulates weakly supervised visual grounding as a query-text matching problem, which can be optimized via the query-based contrastive learning. Based on QueryMatch, we further propose an innovative strategy for effective weakly supervised learning, namely Negative Sample Quality Estimation (NSQE). In particular, NSQE aims to augment negative training samples by actively selecting high-quality query features. Though this strategy, NSQE can greatly benefit the weakly supervised learning of QueryMatch. To validate our approach, we conduct extensive experiments on three benchmark datasets of two grounding tasks, *i.e.,* referring expression comprehension (REC) and segmentation (RES). Experimental results not only show the state-of-art performance of QueryMatch in two tasks, *e.g.,* over +5% IoU@0.5 on RefCOCO in REC and over +20% mIOU on RefCOCO in RES, but also confirm the effectiveness of NSQE in weakly supervised learning. Source codes are available at https://anonymous.4open.science/r/QueryMatch-A82C.

## CCS CONCEPTS

• **Computing methodologies** → **Image segmentation**; **Scene understanding**.

## KEYWORDS

Weakly Supervised Visual Grounding, Contrastive Learning

## 1 INTRODUCTION

Visual Grounding is a significant task in multimedia, which aims to locate the target object in an image based on the natural language descriptions [18, 42, 44, 45]. Compared to conventional detection

Permission to make digital or hard copies of all or part of this work for personal or classroom use is granted without fee provided that copies are not made or distributed for profit or commercial advantage and that copies bear this notice and the full citation on the first page. Copyrights for components of this work owned by others than the author(s) must be honored. Abstracting with credit is permitted. To copy otherwise, or republish, to post on servers or to redistribute to lists, requires prior specific permission and/or a fee. Request permissions from permissions@acm.org.

*ACM MM, 2024, Melbourne, Australia*

© 2024 Copyright held by the owner/author(s). Publication rights licensed to ACM.
ACM ISBN 978-x-xxxx-xxxx-x/YY/MM
https://doi.org/10.1145/nnnnnnn.nnnnnnn

tasks [19, 23, 34, 36], visual grounding not only requires to capture fine-grained object information, but also needs to achieve accurate vision-language alignments. To this end, existing methods often require expensive annotations to learn accurate alignments between natural language descriptions and objects, which is laborious and expensive for practical deployment.

To overcome this limitation, researchers have put their efforts into the weakly supervised learning for visual grounding [7, 8, 14, 15, 22, 24, 26, 38, 39, 41, 46]. Among them, early methods [8, 24–26, 39, 41, 46] often resorts to two-stage object detectors [36] to formulate weakly supervised visual grounding as region-text matching problem. However, these methods are often limited in their inference speed due to the expensive processing of two-stage detectors. Therefore, recent endeavors [14] have been devoted to efficient one-stage modeling for weakly supervised visual grounding. As shown in Fig. 1, they first extract anchor features from a pre-trained one-stage detector, and then conduct anchor-text matching to select the target anchor. Finally, a detection head is used to decode the anchor to the bounding box. This one-stage paradigm does not rely on complex processing like RoI pooling [36], and shows much better efficiency than the two-stage one.

Despite the efficiency, we identify that the anchor-based weakly supervised framework still limits by its object representations, *i.e.,* anchor points. As shown in Fig. 1, anchor points are grid features extracted by the pre-trained detector, *e.g.,* YOLOv3 [34]. Ideally, these anchor points should represent corresponding object informations so that the anchor-text matching can select the target anchor for grounding. However, compared to region features, anchor points are fragmented and struggle to accurately describe object information. Besides, the large receptive field of an anchor often allows it to receive more noisy object informations, which inevitably hinders the accurate vision-language (VL) alignments.

To address these issues, we propose a novel query-text matching framework for one-stage weakly supervised visual grounding, namely QueryMatch. Different from existing methods, QueryMatch resorts to query features extracted from Transformer-based detectors, *e.g.,* DETR [3] and Mask2Former [5], to represent objects. Benefiting from the bipartite matching based pre-training, query features can achieve one-to-one association with visual objects. Therefore, compared to anchor points, query features contain sufficient and accurate object information. By selecting the target query based on the given expression, we can decode the mask or bounding box of the referent by the pre-trained decoder. To enable weakly supervised learning, QueryMatch performs query-text contrastive learning to achieve vision-language alignments via numerous query-text pairs.

Based on QueryMatch, we further propose an innovative strategy for effective weakly supervised learning, namely Negative Sample Quality Estimation (NSQE). As indicated in existing literature [10, 48], the number and quality of negative samples significantly impact

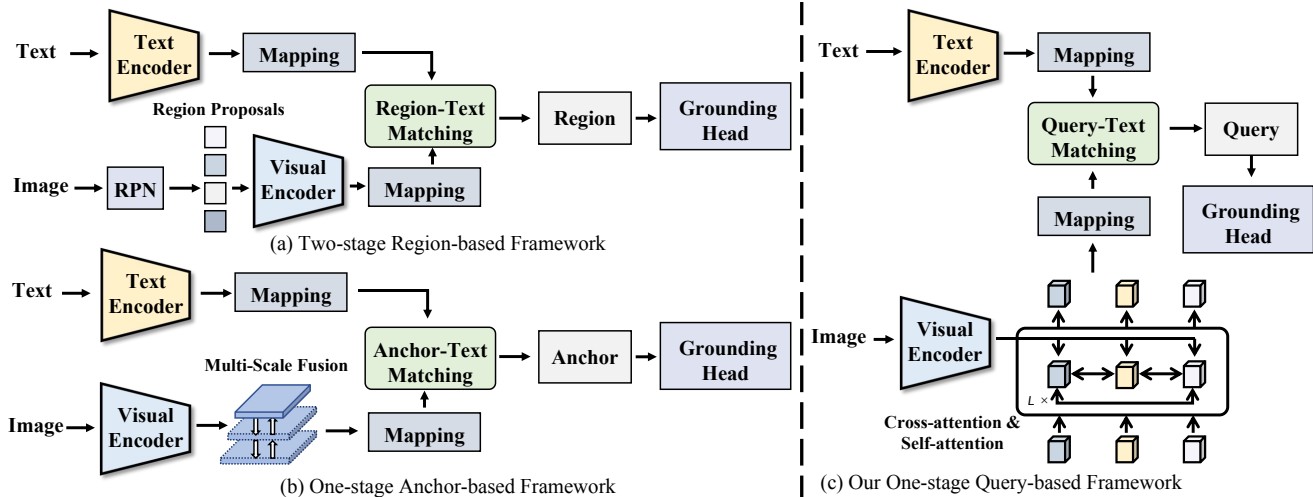

Figure 1: Comparison of different weakly supervised visual grounding schemes.

the effectiveness of contrastive learning. Nevertheless, randomly selected negative samples are often noisy and useless, which will practically hurt the weakly supervised performance. In contrast, NSQE aims to actively select informative samples from a batch for negative query augmentations. Specifically, we design two novel metrics in NSQE to evaluate the quality of negative queries from the uniqueness and the difficulty. By dynamically combining these metrics, we can effectively identify and select high-quality negative queries. With NSQE, we can augment the negative samples by 3 times and achieve significant gains in QueryMatch.

To validate QueryMatch, we conduct extensive experiments in two grounding tasks, namely referring expression comprehension (REC) and segmentation (RES). Experiments on RefCOCO [30], RefCOCO+ [30] and RefCOCOg [28, 30] show that QueryMatch outperforms existing weakly supervised methods by large margins e.g., over +20% for RES on RefCOCO. More importantly, qualitative and quantitative results also confirm the effectiveness of NSQE in query-based contrastive learning.

In summary, this paper makes three main contributions:

- We identify the shortcoming of object representations in existing one-stage weakly supervised visual grounding framework. To overcome this limitation, we propose a novel query-based one-stage framework for weakly supervised visual grounding, namely QueryMatch, which leverages query features to accurately describe object information.
- We propose an innovative strategy for effective weakly supervised learning, namely Negative Sample Quality Estimation (NSQE), which can greatly augment negative samples by selecting high-quality query features.
- The proposed QueryMatch achieves the state-of-the-art results against existing weakly supervised methods on three benchmark datasets of REC and RES.

## 2 RELATED WORK

### 2.1 Weakly Supervised Visual Grounding

Weakly Supervised Visual Grounding includes two tasks, namely weakly supervised REC and weakly supervised RES. These two tasks have attracted considerable attention from numerous researchers recently.

Recent developments in weakly supervised learning for RES can be categorized into two types, depending on whether they utilize text-image pairs as weak supervision. For instance, CTS [7] utilized bounding box annotations for weak supervision, involving a two-stage model training process. Despite achieving promising performance in weakly supervised RES, these methods [7, 17] still incur relatively high annotation costs, particularly on large-scale datasets. Thus, an increasing number of researchers have turned their focus to leverage text-image pairs as a weak supervision signal [15, 22, 38]. For example, Fang Liu et al. employed the CLIP encoder to encode both image and text features. The visual information encoded at the grid level by the image encoder is then matched with textual information, and pseudo labels are generated through a two-stage training process based on these matching results [22]. However, current approaches that depend on weak supervision from text-image pairs demonstrate noticeably inferior performance when compared to other kinds of weakly supervised RES methods.

Meanwhile, Many existing methodologies in weakly supervised REC [8, 24–26, 39, 41, 46] draw inspiration from two-stage supervised REC frameworks, formulating the weakly supervised REC task as a region-to-text ranking problem. The primary challenge within such approaches lies in effectively utilizing supervisory information from image-text pairings. The primary techniques utilized are semantic reconstruction [24, 25, 39] and contrastive learning [8, 14, 46]. Recent efforts in the field of weakly supervised REC have focused on enhancing inference speed by transitioning from resource-intensive two-stage detectors to more efficient one-stage models. These models, exemplified by [14], extract anchor features from pretrained detectors like YOLOv3 [35] and employ anchor-text matching to select target anchors for bounding box decoding. Despite their efficiency, these anchor-based frameworks are hindered by fragmented anchor points' inability to accurately represent object information, leading to challenges in precise vision-language alignments.

Based on these, we propose a novel query-text matching framework for one-stage weakly supervised visual grounding.

**Figure 2: The framework of the proposed QueryMatch. The image and expression are first processed by visual and text encoders. After that, QueryMatch filters out queries with low confidence scores and selects the best-matching one for visual grounding prediction. QueryMatch is weakly supervised with only image-text pairs via a novel query-based contrastive learning paradigm.**

## 2.2 Contrastive Learning

Contrastive learning focuses on minimizing the distances between positive samples while maximizing those between negative samples in the feature space, thereby facilitating the development of effective feature representations. Recently, numerous studies in the multimodal field have embraced contrastive learning [13, 29, 33]. The negative samples in these studies typically comprise real-world instance features, such as images, and it has been observed that the efficacy of contrastive learning tends to improve with an increase in the number of negative samples.

In contrast, the cross-modal contrastive learning framework we propose, based on Query-Text Matching, differentiates itself by employing more granular negative samples compared to those representing real-world entities. For instance, a single image can generate as many as 100 query features. While this approach indeed expands the negative sample pool, it simultaneously introduces a higher degree of informational redundancy within these samples. Notably, we have observed that an excessive increase in the number of negative samples can sometimes lead to a marked decline in performance. One of the key challenges lies in effectively harnessing the rich semantic content these samples offer. To tackle this issue, our research introduces a novel method for estimating the quality of negative samples, aiming to maximize the semantic potential inherent in the negative sample set. As such, it presents innovative contributions to the realm of contrastive learning.

## 3 PRELIMINARY

We first revisit the construction of query features in computer vision. In particular, query-based object detection is a popular paradigm in computer vision. Starting from DETR [3], researchers use a set of learnable vectors to represent candidate objects, also known as queries. These queries are used to interact with image features in different Transformer layers and learn the accurate one-to-one association with objects in images. After that, a decoder layer is adopted to predict masks or bounding boxes based on these queries. During the training phase, the objective of set prediction serves as the loss function. To elaborate, we adopt the architecture of the Mask2former [5]. The features of queries, along with their corresponding position encoding, are treated as learnable vectors, initialized randomly through a standard normal distribution. Subsequently, these vectors undergo sequential processing through the cross-attention layer, self-attention layer, and feed-forward layer within each layer of the Transformer decoder, the process can be formatted as follows:

$$X'_l = \text{LN}(\text{softmax}(M_{l-1} + Q_l \cdot K_{I_l}^T) \cdot V_{I_l} + X_{l-1}), \quad (1)$$

$$\hat{X}_l = \text{LN}(\text{softmax}(\hat{Q}_l \cdot \hat{K}_l^T) \cdot \hat{V}_l + X'_l), \quad (2)$$

$$X_l = \text{LN}(\text{MLP}(\hat{X}_l) + \hat{X}_l), \quad (3)$$

here, $l$ denotes layer index, The set $Q_l \in \mathbb{R}^{N \times C}$ represents the queries at the $l$-th layer, defined as $Q_l = \{q_{l_i}\}_{i=1}^N$, where $N$ signifies the quantity of queries present in each layer. The terms $X'_l$, $\hat{X}_l$, and $X_l \in \mathbb{R}^{N \times C}$ correspond to the outputs of the cross-attention, self-attention, and the feed-forward layers in the $l$-th layer of the Transformer decoder, respectively. These outputs are characterized by $N$ vectors, each of $C$ dimensions. $M_{l-1} \in \{0, -\infty\}^{N \times H_l W_l}$ denotes the attention mask [5], which is equal to 0 when the mask prediction of the previous $l$-th transformer decoder layer is 1, otherwise it is equal to negative infinity. The spatial resolution of image features at the $l$-th layer is represented by $H_l$ and $W_l$. The transformed image features at the $l$-th layer, $K_{I_l}$ and $V_{I_l} \in \mathbb{R}^{H_l W_l \times C}$

are under transformation $f_K$ and $f_V$ respectively. Furthermore, the position encoding is denoted as $P$, leading to the equations $Q_l = f_Q(X_{l-1} + P)$, $\hat{Q}_l = f_{\hat{Q}}(X'_l + P)$, and $\hat{K}_l = f_{\hat{K}}(X'_l + P)$, with the exception of $\hat{V}_l = f_{\hat{V}}(X'_l)$. Here, all $f$ notations stand for linear transformations. Additionally, LN signifies LayerNorm. Considering a Transformer decoder comprising $L$ layers, the final processed set of queries is denoted as $Q = \text{LN}(X_L)$.

After the Transformer decoder processes the queries, the network can decode the grounding information embedded in the corresponding query to obtain the result. This process can be defined as:

$$r_i = \text{Head}(q_i), \tag{4}$$

where $r_i$ denotes the grounding result corresponding to index $i$, and $q_i$ represents the associated query feature. The term Head is used to refer to the layers within the network that are responsible for decoding grounding information from the queries.

## 4 QUERYMATCH

### 4.1 Problem Definition

Given an image $I$ and a textual expression $T$, Visual Grounding aims to locate target objects through bounding boxes or masks. In the context of weakly supervised visual grounding, the text and the corresponding target grounding are unknown during the training stage, posing significant challenges.

Drawing upon the one-to-one correspondence observed between visual grounding results and query features in numerous models utilizing query features for visual grounding [5, 6, 9, 12], and considering the effectiveness of the query feature derived through the Transformer's cross-attention and self-attention mechanisms in capturing intricate relationships between objects in the image, we propose to reformulate the weakly supervised visual grounding problem as a query-text matching problem, which is defined by

$$q^* = \arg\max_{q \in Q} \phi(T, I, q). \tag{5}$$

Here, $q^*$ symbolizes the optimal query, and $Q$ represents the set of queries that have been processed by the transformer decoder within the query-based segmentation model. The notation $\phi(\cdot)$ denotes the mapping module. As delineated in the Section 3, the identification of the optimal query $q^*$ is pivotal. Once $q^*$ is determined, it is possible to decode the query features using subsequent visual grounding layers. This process leads to the derivation of the optimal visual grounding results, expressed as $r^* = \text{Head}(q^*)$.

### 4.2 Query Confidence Selection

The architecture of the QueryMatch framework is depicted in Fig. 2. Prior to the alignment of text with query features, QueryMatch initially selects a subset from the pool of queries to be matched.

Query-based visual grounding model such as DETR [3] and Mask2former [5] usually have many queries do not contribute to predicting the target. To reduce the complexity of cross-modal matching and to economize on matching efforts, the selection is based on the confidence score of each query feature. More precisely, a selected threshold $O$ is established, and only the top $O$ queries with the highest confidence scores are chosen, which is defined by

$$Q_s = \{\text{sort}_{\text{conf}}(\{q_i\}_{i=1}^N)\}_{j=1}^O, \tag{6}$$

where $Q_s$ denotes the queries selected based on confidence, and $\text{sort}_{\text{conf}}$ refers to the process of descending sorting according to the confidence scores of the query features.

Similar to the image-contrastive learning framework, e.g., CLIP [33], QueryMatch likewise projects both visual and textual features into a shared semantic space, facilitating the learning of vision-language alignment.

### 4.3 Query-based Contrastive Learning

To enable the model to learn how to identify the optimal query that aligns with the text through weak supervision information, thereby achieving visual grounding, we introduce a query-based contrastive learning framework. Specifically, we begin by encoding features of the input images and text to obtain visual features and text features, denoted as $v \in \mathbb{R}^{H \times W \times C}$ and $t \in \mathbb{R}^C$, respectively. The query interacts with the visual features via the transformer decoder, resulting in query features containing the necessary semantic information for visual grounding, denoted as $Q \in \mathbb{R}^{N \times C}$. We then perform confidence-based selection on $Q$ to obtain $Q_s \in \mathbb{R}^{L \times C}$, where $Q_s = \{q_i\}_{i=1}^O$.

Finally, we map the selected query $q_i$ and the text feature $t$ linearly onto the same semantic space, written by:

$$f_{q_i} = q_i \cdot W_q + b_q, \tag{7}$$

$$f_t = t \cdot W_t + b_t, \tag{8}$$

where $W_q$ and $W_t$ are the projection matrices, and $b_q$ and $b_t$ are biases.

In visual grounding, a natural language description typically corresponds to a target in the image. Theoretically, one query acts as the positive example, while the rest serve as negative ones. We select the query with the highest matching score to the current text in the image as the positive sample. Negative queries are chosen from queries in images not matching the current text. Our objective is to maximize the similarity between the text and the matching query and minimize the similarity with unmatched queries. Therefore, we define the contrastive loss as follows:

$$\mathcal{L}_c = -\log \frac{\exp\left(\text{sim}(f_{q_1}^i, f_t^i)/\tau\right)}{\sum_{n=1}^K \sum_{j=1}^B \mathbb{I}_{\neg(i=j \,\wedge\, n \neq 1)} \exp\left(\text{sim}(f_{q_n}^j, f_t^i)/\tau\right)}, \tag{9}$$

here, $f_{q_n}^j$ represents the positive and negative queries sampled from a batch, with $f_{q_1}^i$ being the positive one. The function $\text{sim}(\cdot)$ is utilized for calculating similarity, which is defined by the inner product. The term $\mathbb{I}_{\neg(i=j \,\wedge\, n \neq 1)}$ is the indicator function, equal to 0 when $i = j$ and $n \neq 1$. $K$ is the number of negative queries selected from each unmatched image-text pair, $B$ is the batch size, and $\tau$ is the temperature parameter in Hinton et al. [11].

## 5 NEGATIVE SAMPLE QUALITY ESTIMATION

In this section, we detail a negative sample selection strategy aimed at leveraging the semantic information embedded in image-text pairs more effectively to further enhance the performance.

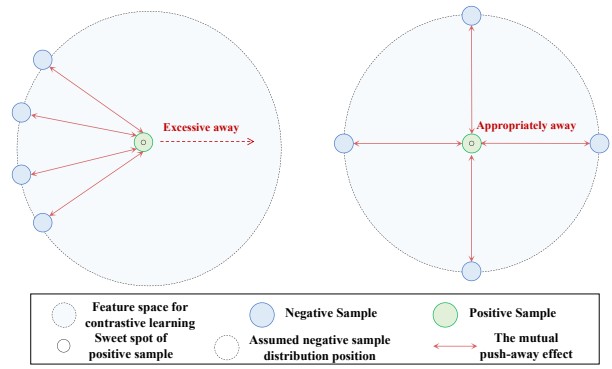

**Figure 3: The importance of ensuring specific uniqueness in the feature space for negative samples.**

Driven by the observation that the mere quantity of negative samples does not guarantee superiority, we propose to estimate their quality. The quality is determined by two factors: distinguishability and uniqueness in the semantic space. In particular, a sample that significantly challenges distinction and exhibits a high degree of uniqueness in the semantic space is considered to be of superior quality. We formulate the quantitative evaluation by:

$$S_{u_i} = \text{Norm}\left(-\max_{j=1}^{m} \cos\left(f_{q_i}, f_{q_j^n}\right)\right), \quad (10)$$

$$S_{d_i} = \text{Norm}\left(\text{sim}(f_{q_i}, f_t)\right), \quad (11)$$

where $S_{u_i}$ denotes the uniqueness of the negative samples, and $S_{d_i}$ denotes the difficulty in discrimination. The term $f_{q_i}$ represents the negative query under evaluation for quality, and $f_{q_j^n}$ refers to a previously selected negative query. The variable $m$ indicates the number of negative queries selected in the current iteration. The symbol cos denotes the cosine similarity calculation, and Norm signifies the min-max normalization.

To elucidate, our fundamental principle asserts that cosine similarity effectively captures the resemblance among negative samples in the feature space. When a negative sample has low similarity with its most similar negative sample in this space, it indicates greater uniqueness. Additionally, higher similarity between negative samples and the current input text makes them more challenging to distinguish.

To introduce the final strategy for quality estimation, it is crucial to highlight the significance of uniqueness in negative samples within the feature space, as depicted in Fig.3. Excessive clustering of a specific category of negative samples within this space can unintentionally displace positive samples from their ideal locations. This phenomenon can be understood intuitively: an overrepresentation of a single type of negative sample can reduce the relative significance of other negative samples, leading to skewed results. In contrast, a more balanced representation of negative samples contributes to enhanced outcomes.

In conclusion, to guarantee that negative samples maintain adequate levels of uniqueness and challenge in discrimination, we propose a straightforward and effective method for estimating their quality, defined by:

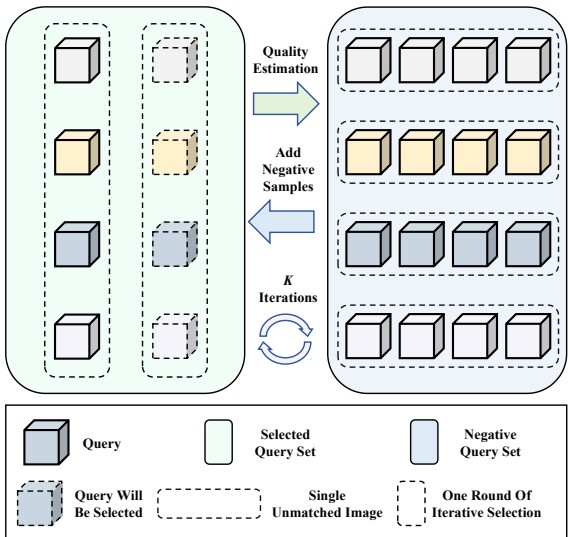

**Figure 4: Schematic diagram illustrating high-quality negative sample selection and update process.**

$$S_{q_i} = S_{u_i} \cdot S_{d_i}, \quad (12)$$

where $S_{q_i}$ denotes the quality score of the negative sample $f_{q_i}$. By estimating the quality scores for each negative sample, we can rank them in descending order, enabling the selection of a appropriate number of negative samples for contrastive learning.

As illustrated in Fig. 4, to fully utilize the information present in image-text pairs and to ensure equitable sampling of negative samples across each image-text pair, we conduct our sampling based on the number $K$ of negative sample selections for each unmatched image-text pair. It is important to note that the uniqueness of negative samples is dynamic and evolves with the selection of different samples. Initially, when no negative sample is chosen, the uniqueness value $S_u$ is set to 1. Consequently, the quality scores of negative samples are dynamically updated throughout the selection process. This process involves $K$ iterative steps. In each step, after selecting a negative sample from every mismatched image-text pair, we reassess and update the quality scores of the remaining negative samples based on their revised uniqueness.

$$f_{q^n} \longleftarrow \underset{q \in Q}{\arg\max}(f_q(S_q^i)), \quad (13)$$

where $i$ indicates the $i$-th iteration step. The term $f_{q^n}$ denotes the set of negative samples selected up to the current iteration, and $S_q^i$ represents the quality score associated with the current negative samples.

In addition, the variable $n_{gpu}$ denotes the number of GPUs utilized in distributed training, and $B$ represents the batch size. Then we can mathematically articulate the relationship between the total number $M$ of negative samples chosen and the sampling unit $K$ by:

$$M = (B - n_{gpu}) \cdot K. \quad (14)$$

Table 1: Comparisons with state-of-the-art methods on three RES benchmark datasets. [1]

| Method | Sup. | RefCOCO | | | RefCOCO+ | | | RefCOCOg |
|---|---|---|---|---|---|---|---|---|
| | | val | testA | testB | val | testA | testB | val-g |
| CTS [7] | $\mathcal{B}+\mathcal{T}$ | 58.01 | 60.52 | 55.48 | 47.12 | 50.86 | 40.26 | 46.03 |
| PKS [17] | $\mathcal{C}+\mathcal{T}$ | 49.27 | 52.23 | 45.64 | 37.79 | 42.09 | 32.87 | 36.43 |
| AMR† [32] | $\mathcal{T}$ | 14.12 | 11.69 | 17.47 | 14.13 | 11.47 | 18.13 | 15.83 |
| GroupViT† [43] | $\mathcal{T}$ | 18.03 | 18.13 | 19.33 | 18.15 | 17.65 | 19.53 | 19.97 |
| CLIP-ES† [21] | $\mathcal{T}$ | 13.79 | 15.23 | 12.87 | 14.57 | 16.01 | 13.53 | 14.16 |
| GbS† [1] | $\mathcal{T}$ | 14.59 | 14.60 | 14.97 | 14.49 | 14.49 | 15.77 | 14.21 |
| WWbL† [37] | $\mathcal{T}$ | 18.26 | 17.37 | 19.90 | 19.85 | 18.70 | 21.64 | 21.84 |
| TRIS [22] | $\mathcal{T}$ | 31.17 | 32.43 | 29.56 | 30.90 | 30.42 | 30.80 | 36.00 |
| QueryMatch (ours) | $\mathcal{T}$ | **59.10** | **59.08** | **58.82** | **39.87** | **41.44** | **37.22** | **43.06** |

Table 2: Comparisons with state-of-the-art methods on three REC benchmark datasets.

| Method | RefCOCO | | | RefCOCO+ | | | RefCOCOg | Inference |
|---|---|---|---|---|---|---|---|---|
| | val | testA | testB | val | testA | testB | val-g | Speed |
| VC[31]CVPR18 | - | 32.68 | 27.22 | - | 34.68 | 28.10 | 29.65 | - |
| KAC Net[4]CVPR18 | - | - | - | - | - | - | - | - |
| MATN[47]CVPR18 | - | - | - | - | - | - | - | - |
| ARN[24]ICCV19 | 32.17 | 35.25 | 30.28 | 32.78 | 34.35 | 32.13 | 33.09 | 5.7fps |
| IGN[46]NeurIPS20 | 34.78 | 37.64 | 32.59 | 34.29 | 36.91 | 33.56 | 34.92 | - |
| DTWREG[39]TPAMI21 | 38.35 | 39.51 | 37.01 | 38.91 | 39.91 | 37.09 | 42.54 | 5.9fps |
| RelR[26]CVPR21 | - | - | - | - | - | - | - | - |
| NCE+Distillation[41]CVPR21 | - | - | - | - | - | - | - | - |
| RefCLIP [14]CVPR23 | 60.36 | 58.58 | 57.13 | 40.39 | 40.45 | 38.86 | 47.87 | 31.3fps |
| QueryMatch (ours) | **66.02** | **66.00** | **65.48** | **44.76** | **46.72** | **41.50** | **48.47** | 17.7fps |

# 6 EXPERIMENTS

## 6.1 Datasets and Metric

**Datasets. RefCOCO** [30] consists of 142,210 referring expressions and 50,000 objects extracted from 19,994 MSCOCO [20] images. The expressions in RefCOCO predominantly pertain to absolute spatial information. **RefCOCO+** [30] comprises 141,564 referring expressions corresponding to 49,856 bounding boxes from 19,992 MSCOCO images. RefCOCO+ shares the same data splits as Ref-COCO, but its descriptions focus on relative spatial information and appearance, such as color and texture. **RefCOCOg** [28, 30] contains 104,560 referring expressions associated with 54,822 bounding boxes in 26,711 images. In comparison to RefCOCO and RefCOCO+, RefCOCOg exhibits longer and more complex expressions. In our experiments, we utilize the *google* split [28] of RefCOCOg.

**Metric.** Consistent with [14, 22], we utilize IoU@0.5 as the metric for REC, indicating a correct prediction when the IoU between the predicted and ground-truth boxes exceeds 0.5. Additionally, for evaluating RES accuracy, we employ mask Intersection-over-Union (IoU) and Prec@0.5 (P@0.5).

## 6.2 Implementation Details

In the QueryMatch framework, we employ a pretrained Mask2former, featuring a Swin-B backbone [5, 27], to extract Query features. This

Mask2former is pretrained on the MS-COCO dataset [20], with images from the *val* and *test* sets excluded for all three datasets under consideration. The language encoding is accomplished using a bidirectional GRU [2], which is further augmented by a self-attention layer [40]. Consistent with the approach in [14], Input images are resized to dimensions of $416 \times 416$. The training regimen is executed on two 24G Nvidia RTX 4090 GPUs, with an allocated batch size of 16 for each GPU. During this phase, the parameters of the Mask2former are kept frozen. For the purpose of Query-based contrastive learning, the linear projection is also set to a dimension of 512 for most datasets, and by default, 20 queries with the highest confidence scores per image are selected. Model training is conducted using the *Adam* optimizer [16] with a constant learning rate of 1e-4. The number of training epochs is set at 25.

## 6.3 Quantitative Analysis

**1.Comparison to the state-of-the-arts.**

As shown in Tab.1 and Tab.2, QueryMatch significantly outperforms current state-of-the-art methods on both weakly supervised

---

[1]Type indicates what kinds of visual features the approach is based on to match text. Sup. denotes the supervision type ($\mathcal{B}$: box-level labels, $\mathcal{C}$: click-level annotations, $\mathcal{T}$: text description labels). † indicates the methods adapted from other related tasks by the related paper [22]. "-" denotes unavailable values.

**Table 3: Ablation of the confidence selection threshold on RefCOCO.**

| $O$ | RES | | | REC | | |
|---|---|---|---|---|---|---|
| | val | testA | testB | val | testA | testB |
| 100 | 57.02 | 55.35 | 58.06 | 63.44 | 61.57 | 64.06 |
| 50 | 57.50 | 55.82 | 58.29 | 63.95 | 62.06 | 64.69 |
| **20** | **57.82** | 56.54 | **58.43** | **64.48** | 63.02 | **64.79** |
| 10 | 56.19 | **56.60** | 54.86 | 62.54 | **63.16** | 60.79 |

**Table 4: Ablation of negative sample quality estimation strategy on RefCOCO.**

| Strategy | $M$ | RES | | | | REC | | | |
|---|---|---|---|---|---|---|---|---|---|
| | | val | testA | testB | Avg. | val | testA | testB | Avg. |
| random | 30 | 35.70 | 36.42 | 35.92 | 36.01 | 40.01 | 41.01 | 40.00 | 40.34 |
| difficult | 30 | **57.82** | **56.54** | **58.43** | **57.60** | **64.68** | **63.02** | **64.79** | **64.16** |
| NSQE | 30 | **57.82** | **56.54** | **58.43** | **57.60** | **64.68** | **63.02** | **64.79** | **64.16** |
| random | 90 | 37.01 | 37.48 | 37.39 | 37.29 | 41.55 | 42.27 | 41.45 | 41.76 |
| difficult | 90 | 55.02 | 53.93 | 56.63 | 55.19 | 61.69 | 60.08 | 63.32 | 61.70 |
| NSQE | 90 | **58.72** | **59.11** | **58.81** | **58.88** | **65.70** | **66.13** | **65.46** | **65.76** |
| random | 120 | 37.65 | 38.09 | 38.31 | 38.02 | 42.30 | 42.96 | 42.57 | 42.61 |
| difficult | 120 | 54.72 | 52.85 | 57.04 | 54.87 | 61.24 | 59.31 | 63.85 | 61.47 |
| NSQE | 120 | **59.10** | **59.08** | **58.82** | **59.00** | **66.02** | **66.00** | **65.48** | **65.83** |
| random | 180 | 39.50 | 39.11 | 40.39 | 39.67 | 44.40 | 44.16 | 44.89 | 44.48 |
| difficult | 180 | 50.35 | 48.61 | 52.17 | 50.38 | 56.27 | 54.57 | 58.04 | 56.20 |
| NSQE | 180 | **58.15** | **58.00** | **58.49** | **58.21** | **65.18** | **64.86** | **65.18** | **65.07** |

RES and REC tasks. This fully demonstrates the cross-modal alignment capabilities of QueryMatch. As shown in Tab.1, the comparison models were trained using weakly supervised information such as bounding box annotations, click annotations, and text. Among the existing weakly supervised RES models trained on image-text pairs, TRIS, which employs a pre-trained CLIP encoder and a two-stage pseudo-label generation process for training, previously held the highest performance. However, our QueryMatch significantly outperforms TRIS across all datasets through its novel Query-Text matching framework. For instance, on the RefCOCO dataset, accuracy improved by more than 20%. As can also be seen from Tab.2, QueryMatch surpasses the previous state-of-the-art weakly supervised REC methods by more than 5% on the RefCOCO dataset.

Moreover, QueryMatch substantially narrows the performance gap between image-text pair-based RES models and CTS as well as PKS models that utilize more granular weak supervision information like bounding box and click annotations. Compared to the CTS model on the RefCOCO datasets, it improves accuracy by an average of 1%, and notably surpasses the overall performance of the PKS model.

**2.Ablation of QueryMatch**

We began by evaluating the query confidence selection strategy. In this scenario, the negative sample selection strategy selected the most difficult-to-distinguish query for each unmatched image-text pair. As indicated in Tab.3, implementing the query confidence selection resulted in enhanced outcomes. This approach not only improved computational efficiency but also achieved better accuracy, specifically an average increase of 0.79% on the RefCOCO

**Table 5: Ablation of the formula for negative sample quality estimation on RefCOCO.**

| Formula | RES | | | REC | | |
|---|---|---|---|---|---|---|
| | val | testA | testB | val | testA | testB |
| $S_{d_i}$ | 54.72 | 52.85 | 57.04 | 61.24 | 59.31 | 63.85 |
| $S_{u_i}$ | 31.85 | 34.23 | 30.96 | 39.82 | 42.67 | 39.14 |
| $S_{u_i}^{\dagger}$ | 58.64 | **59.22** | 58.68 | 65.53 | **66.10** | 65.16 |
| $S_{u_i} + S_{d_i}$ | 58.64 | 58.54 | 58.63 | 65.45 | 65.46 | 65.12 |
| $S_{u_i} \cdot S_{d_i}$ | **59.10** | 59.08 | **58.82** | **66.02** | 66.00 | **65.48** |

dataset. Therefore, in subsequent experiments, we opted for a selection number threshold of $O = 20$.

Concerning the negative sample selection strategy, our initial approach involved random selection among the queries that did not match the image-text pairs. As Tab.4 illustrates, this method sustained low accuracy and exhibited a significant decline when the selection number exceeded 150. Another intuitive strategy was to select a specific number of negative samples from mismatched image-text pairs based on the difficulty of distinguishing them. However, as Tab.4 demonstrates, although this approach generally more effective than random selection, it started to diminish significantly beyond 90 negative sample selections. At 180 selections, there was a 7.22% drop in accuracy compared to the initial 30 selections. This trend suggests that the strategy focusing only on the difficulty of distinguishing has a limited capacity to utilize image-text information effectively.

To optimize the utilization of image-text pairs, we proposed a selection strategy that integrates the uniqueness of negative samples with their discrimination difficulty. The results reveal that our method significantly improved accuracy compared to the previous strategies, demonstrating stronger robustness and scalability. Across a range of 30 to 180 negative samples, accuracy remained consistently high. The peak average accuracy, 59% on RefCOCO, was achieved with 120 negative samples. Even with 180 selections, the accuracy remained stable at 58.21%, which is 7.83% higher than the strategy focusing only on discrimination difficulty.

Finally, we compared our approach with methods based only on the uniqueness of negative samples and using the sum of uniqueness and discrimination difficulty as a quality measure. As shown in Tab.5, our method exhibited the highest accuracy, substantially outperforming strategies that considered only the difficulty of discrimination. This underscores the importance of negative sample feature distribution in contrastive learning. It's noteworthy that the strategy $S_{u_i}^{\dagger}$ only considers the uniqueness of negative samples, presupposes selecting the most challenging negative sample in the first iteration step. Opting for random selection in the first iteration step $S_{u_i}$ significantly reduces accuracy, highlighting the critical importance of both metrics in evaluating the quality of negative samples. Additionally, in other datasets, our strategy consistently outperforms, exhibiting a more significant improvement over the $S_{u_i}^{\dagger}$ strategy than Tab.5 indicates.

## 6.4 Qualitative Analysis

To provide an intuitive understanding of the efficacy of our proposed method, we conducted visualizations, as illustrated in Fig.5

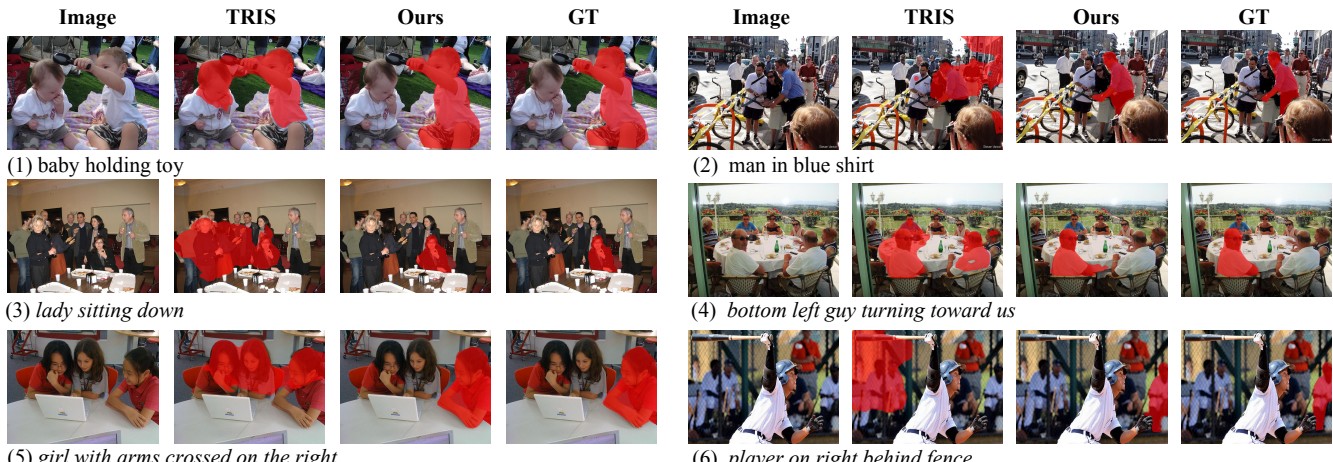

**Figure 5: Visualized predictions of our approach and the state-of-the-art method [22] leveraging image-text pairs for weakly supervised training: The red mask indicates segmentation results, and GT denotes ground truth segmentation results.**

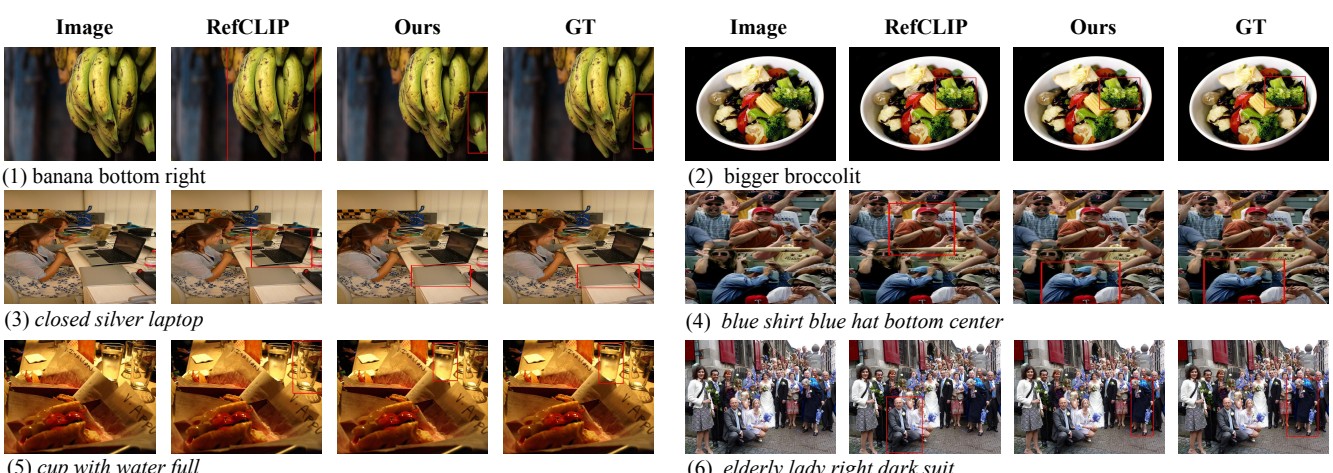

**Figure 6: Visualized predictions of our approach and the state-of-the-art method [14] employing anchor-text matching framework: The red mask indicates detection results, and GT denotes ground truth detection results.**

and Fig.6. Compared to the previous state-of-the-art models, the weakly supervised RES model TRIS and the weakly supervised REC model RefCLIP, our approach demonstrates substantial improvements. Evidently, TRIS often struggles with accurately locating targets, leading to significant prediction errors in many cases. Meanwhile, RefCLIP tends to make localization errors in scenes where the relationships between objects are complex. In contrast, QueryMatch exhibits a high degree of precision in target localization in most scenarios. This accuracy persists even in complex situations, such as when objects are intricately interrelated or densely packed. Our method's ability to consistently deliver accurate predictions under such challenging conditions highlights its robustness and effectiveness.

## 7 CONCLUSIONS

In this paper, we present QueryMatch, a novel weakly supervised visual grounding framework that only relies on image-text pairs. The framework reformulates visual grounding as a Query-Text matching problem, notably simplifying the training complexity by employing an end-to-end paradigm. Further, inspired by experimental observations that highlight the crucial role of negative sample quality in cross-modal contrastive learning, we propose a strategy, namely NSQE, to estimate the quality of negative samples. This strategy significantly boosts performance by selecting high-quality negative samples, emphasizing their uniqueness and difficulty in discrimination. Experimental results validate the effectiveness and robustness of both the Query-Text matching framework and the negative sample quality estimation strategy.

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
