# OpenReview forum: "QueryMatch: A Query-based Contrastive Learning Framework for Weakly Supervised Visual Grounding"
_acmmm.org/ACMMM/2024/Conference — MM2024 Poster_

### Official Review · Reviewer_7tWr · 2024-05-22

**Rating:** 4
**Confidence:** 4

**Summary:**

The paper introduces a weakly supervised visual grounding method, which is based on the transformer backbone and conducts the contrastive learning by the learnable query. Additionally, the authors propose a negative sample selection strategy that enhances the quality of negative samples. The approach achieves good performance on both the RES and REC tasks.

**Strengths:**

1) The method improves the strong transformer-based method from full supervised into the weakly supervised manner, which help for the future work.
2) The quality estimation guarantees that the negative sample is orthogonal, and the ablation experiments results are encouraging.

**Limitations:**

1) As I know for the weakly supervised visual grounding task, the input are the text and the image without grounding box, so the text is known towards models. However, in the line 372, you claim the text is unknown during training but you provide the text input. Can you explain it?
2) I can understand the running process, but the Figure 2 is confusion for me, especially the relation between the green box and blue box. For example, which module's parameters have been updated by contrastive learning?
3) The experiments only conducted on one backbone cannot demonstrate method generalization.
4) Is the method perform well in the full supervised manner?
Typo Errors
Two comma in Line 412.

**Suitability:**

2

---

### Official Review · Reviewer_EAUW · 2024-05-24

**Rating:** 6
**Confidence:** 3

**Summary:**

The paper introduces QueryMatch, a query-based one-stage framework for weakly supervised visual grounding that improves vision-language alignment by using query features to represent candidate objects. It redefines the task as a query-text matching problem optimized through query-based contrastive learning. The authors also propose Negative Sample Quality Estimation (NSQE) to enhance learning by selecting high-quality negative samples. Experiments on benchmark datasets show QueryMatch's state-of-the-art performance and confirm NSQE's effectiveness.

**Strengths:**

1. The motivation is clear, and the paper is well-written and easy to follow.
2. The novelty of this paper is significant, which is clearly shown in Figure 1. Authors take a deep look into existing works and re-formulates weakly supervised visual grounding tasks as a query-text matching problem, which seems interesting and inspiring for me.
3. The proposed method achieves the state-of-the-art performance on benchmark datasets with relatively smaller inference costs. Both quantitative and qualitative analyses are very detailed and persuasive.

**Limitations:**

None in particular

**Suitability:**

3

---

### Official Review · Reviewer_i7XU · 2024-05-24

**Rating:** 4
**Confidence:** 4

**Summary:**

The paper titled "QueryMatch: A Query-based Contrastive Learning Framework for Weakly Supervised Visual Grounding" proposes a novel query-based framework for weakly supervised visual grounding. The authors identify the limitations of existing one-stage anchor-based frameworks and introduce query features to accurately describe object information. They also propose a negative sample quality estimation strategy to select high-quality negative samples for contrastive learning. Extensive experiments on benchmark datasets demonstrate the effectiveness of the proposed framework and strategy.

**Strengths:**

(1) The paper introduces a novel query-based framework for weakly supervised visual grounding, which addresses the limitations of existing anchor-based frameworks and improves the accuracy of vision-language alignments.
(2) The proposed negative sample quality estimation strategy effectively selects high-quality negative samples, leading to improved performance in weakly supervised learning.
(3) The paper provides extensive experiments on benchmark datasets and achieves state-of-the-art results in both referring expression comprehension (REC) and segmentation (RES) tasks.

**Limitations:**

(1) The paper lacks a detailed description of the implementation details, such as the specific architecture of the proposed framework and the hyperparameter settings.
(2) The paper does not provide a thorough comparison with widely-known baselines in the field, which makes it difficult to assess the novelty and significance of the proposed framework.
(3) The paper lacks a comprehensive analysis of the limitations and potential future directions of the proposed framework.

Overall, the paper presents a novel query-based framework for weakly supervised visual grounding and introduces an effective strategy for selecting high-quality negative samples. The experimental results demonstrate the superiority of the proposed framework over existing methods. However, more details and comparisons with baselines would further strengthen the paper.

**Suitability:**

2

---

### Official Review · Reviewer_CGev · 2024-05-28

**Rating:** 2
**Confidence:** 3

**Summary:**

The author proposed QueryMatch, a framework target for weakly supervised visual grounding(WSVG). The author reformulates WSVG as a query-text matching problem and optimizes the problem using query-based contrastive learning. To solve the problem, the author proposed NSQE, which can select high-quality negative samples. Experiment results show the effectiveness of the proposed framework.

**Strengths:**

1. The experiment shows a huge improvement in the RES problem.
2. The paper is well-organized and easy to follow.

**Limitations:**

1. In Table 2, the author missed lots of results for other methods. Also, the RefCLIP has better results in their paper, so why not compare with that result?
2. In Table 1, why your result is better than PKS or CTS which provides more information, it should be the upper bound for RES. The author should provide more analysis of why their method can have such a high performance or provide an experiment that also provides C+T or B+T to provide more comprehensive results.
3. I feel confused about the evaluation metrics that the author used. The author claims that for RES, they use two different evaluation metrics(IoU, P@0.5), but there is only one result.

**Suitability:**

3

---

### Meta-Review · Area_Chair_z1Pi · 2024-07-07

**Recommendation:** Accept (Poster)
**Confidence:** 5

**Metareview:**

This paper introduces QueryMatch, a novel query-based framework for weakly supervised visual grounding that addresses the limitations of noisy anchor representations. QueryMatch uses query features for accurate one-to-one object associations and employs query-based contrastive learning. It also introduces Negative Sample Quality Estimation (NSQE) to enhance training with high-quality negative samples. Extensive experiments on benchmark datasets demonstrate QueryMatch's state-of-the-art performance and the effectiveness of NSQE in weakly supervised learning.

Pros
- This paper introduces a novel query-based framework for weakly supervised visual grounding, addressing limitations of existing anchor-based frameworks and improving vision-language alignments. The proposed method effectively selects high-quality negative samples, leading to improved performance in weakly supervised learning.
- Comprehensive experiments on benchmark datasets, achieving state-of-the-art results in both referring expression comprehension (REC) and segmentation (RES) tasks. In particular, the results show a substantial improvement in the RES problem.
- The paper is well-organized and easy to follow.

Cons
- There are flaws in the experiments, such as missing results for many methods in Table 2 and no comparison with RefCLIP. Table 1 results exceed expected upper bounds (PKS or CTS) without adequate explanation. There is also an insufficient comparison with well-known baselines and a lack of detailed architecture and hyperparameter settings.
- There are some typos too as revealed by the reviewers.

Overall, this is a decent paper with a novel idea and solid experiments. All reviewers lean towards accepting this paper. The authors are recommended to address all reviewer comments in the camera-ready version.